



# Simulating Carbon Sequestration using Cellular Automata and land use assessment; Case of: Karaj City, Iran

**Ali Khatibi[1], Sharareh Pourebrahim [1], Mazlin Bin Mokhtar[2]**

1–Department of Environmental Sciences, Faculty of Natural Resources, University of Tehran, Iran sh_pourebrahim@ ut.ac.ir
2-Institute for Environment and Development (LESTARI), National University Malaysia(UKM), 43300, Malaysia

**Abstract**

In this study, in the city of Karaj five classes of land use-cover including residential, agriculture, rangeland, forest and barren areas were considered and randomly in each class a total of 20 points were selected and vegetation and soil samples were taken. In plant samples, the amount of carbon sequestration was determined by calculating the amount of organic carbon by dry weight and in soil samples, the amount of carbon sequestration was determined by using Walleky and Black method, too.  For each area, the average value of carbon sequestration of samples was introduced as 'carbon sequestration index' of that class. Average values for each category were determined as an indicator of carbon sequestration of that class and then by using the DINAMICA EGO software a simulation was conducted using cellular automata approach to simulate changes in the classes of land use-cover in the city of Karaj. Finally, by using carbon sequestration index and the results of the simulation, changes in carbon sequestration in each class were calculated. On this basis, it was found that in the 15-year period from 2014 to 2029, not considering the residential class as the effective use of carbon sequestration, the greatest amount of carbon sequestration was found in the agricultural class and the lowest carbon sequestration was found in barren area. Also, agriculture class will be faced with the huge reduction of carbon sequestration, because of expansion of the residential area.

**Keywords:** Carbon Sequestration, Land use-cover Changes, Cellular Automata, City of Karaj




## 1  Introduction

Carbon sequestration can be defined as the process of removing carbon from the atmosphere and depositing it in a reservoir (UNFCCC, 2015). Carbon sequestration describes long-term storage of carbon dioxide or other forms of carbon to either mitigate or defer global warming and avoid dangerous climate change. It has been proposed as a way to slow the atmospheric and marine accumulation of greenhouse gases, which are released by burning fossil fuels  (Hodrien, 2008). Terrestrial carbon sequestration is the result of a balance between the different stages of the carbon cycle in the biosphere and pedosphere, such as photosynthesis, plant growth, congestion and carbon accumulation in soils and carbon emissions from breathing organisms, microbial decomposition of leaf litter, and oxidation of organic carbon in soil and land degradation. Several factors are involved in this process, which are classified in two categories of physical and managerial factors. Physical factors are divided into three categories of factors: soil, climate and terrain geometry. These factors are difficult and sometimes impossible to control. Therefore, in the management of carbon sequestration, only managerial factors are in the control of humans (Ferreiro *et al.*, 2010).

Because the majority of carbon sequestration is in the soil, organic carbon of the soil plays an important role in the process of carbon sequestration. The amount of carbon in the soil significantly changes according to location, topography, bedrock or vegetation type and previous management approaches. Also time-wise, in the growing season and the time of decomposition processes, the amount of carbon found in the roots, litter and biomass of microorganisms in the soil can vary. The main processes of carbon sequestration in soil includes humic organic material, transformation of humus into organic-inorganic compounds in soil, placing these compounds deep in the soil in areas that are plowed, deep root plants and calcareous soil. Instead, processes that reduce the amount of organic matter in soil include erosion, compaction and reduction of soil permeability, loss of soil structure, mineralization and oxidation of humic substances (Bruce *et al.*, 1999).

Nowadays, with the development of vegetation covers of hard wood trees and shrubs, this method is now used more effectively than other methods to reduce carbon dioxide in the air. Although the amount and rate of carbon sequestration is higher in mild and humid tropical forest ecosystems however, the speed of the chemical analysis and biological processes that cause the release of carbon dioxide, due to high humidity and high ambient temperature is also very high. Therefore, arid and semi-arid regions are the best options for carbon sequestration. The increase in biomass of hard wood plants in these regions has many advantages due to the reduction of carbon sequestration (Lal, 2008). That is why international organizations such as FAO and UNDP have chosen these regions to implement programs of carbon sequestration to reduce greenhouse gases (Ghanbari, 2014).

Through photosynthesis, plants absorb carbon and then return some of it to the atmosphere through respiration. The carbon that remains in plant tissue either gets consumed by the animals or gets added to the soil as litter (when the plants decompose and die). In the first way, the carbon gets stored in the soil and is considered as soil organic matter. Soil organic matter is a complex mixture of carbon syntheses including decomposing plant and animal tissues, microbes (protozoa, nematodes, fungi and bacteria) and carbon minerals in the soil. Carbon can remain stored in the soil for thousands of years. Factors such as climate, natural vegetation, soil texture and drainage can influence the amount and duration of carbon storage in soil (Schlesinger, 1984). The land use-cover is divided into multiple classes. This classification includes sorting the objects in groups or collections of objects based on the relationships that exist between them



(Sokal, 1974; Anderson *et al.*, 1976; Gregorio & Jansen, 2005; Şatır & Berberoğlu, 2012). The changes in land use-cover can be caused by natural factors or human's actions. Although people have always changed the land for their basic needs in life, the speed of these changes is already much higher than in the past, which led to many changes in the environment and ecological processes at the local, regional and global spectrum. These changes in land, causes some changes in the climate, the amount of pollution and biodiversity. Monitoring and modeling historical situation of the region can be used to anticipate the negative effects of these changes in land use-cover, in order to protect resources, effectively. As a result, finding appropriate methods for modeling the land use-cover changes is a global goal for researchers and politicians (Ellis & Pontius, 2010).

One of the most popular method to simulate spatial changes is cellular automata. A cellular automata is a system of a finite number of cells that are located next to each other in terms of location and continuity and are stored together in a regular grid of cells in the form of a raster image. This model is well suited to simulate phenomenon of spatial landscape (Al-ghamdi, 2012; Guan & Clarke, 2010; Jantz *et al.*, 2010; KwadwoNti, 2013).

**Materials and Methods**

After selection of city of Karaj as the study area and characterization of sampling points, plant and soil samples were taken from each and every point. Sampling at any point was usually done in a season when plant growth in that area had been completed. Which means, depending on the coverage of each class, and depending on the type of plant species in each class their sampling time was determined (Karimian, 2009). For samples of grass and shrub cover, it was sure to sample the aerial and underground biomasses separately. For measuring and sampling the aerial biomass direct measurement method was used. For this purpose, the aerial biomass to the ground was cut (completely cutting the aerial parts) and after drying them in the laboratory, weighed, and then by including plant basis weight per unit area, weight of aerial biomass and total biomass per hectare were estimated (Mesdaghi, 2001).

In the sampling points, plots with specific measures ($1 \times 1$ square m) had been placed. Litter in each plot was harvested as part of the aerial biomass. Litter samples were transported to the laboratory and after getting fully dried, the dry weight of each sample was recorded separately and then the total weight of litter per square meter was determined. Also, belowground biomass of the sampled species were extracted. Usually these profiles had the dimensions of $30 \times 30 \times 30$ cubic cm. In each plot, the roots were collected and after drying, weighted, and then by calculating the root weight per unit area the amount of studied factors were determined. In forest cover, usually a number of pure species with an average age of 20 years were selected. In order to determine the weight of biomass of standing trees, depending on forest density and homogeneity of mass, plots of different sizes were used. To do this, Plots were randomly located in forest stand and the number of bases of the tree in each were counted and then we attempted to measure the height and diameter at breast to determine the volume of the tree. To determine the bulk density of the tree, we took samples from a tree branch in each stand. For this purpose, a piece of tree branches with specified dimensions were cut and then their dry weight and volume were measured. By dividing weight by volume, bulk density of the tree was determined. Then, by multiplying the volume of trees in bulk density, biomass weight per unit area was determined. The method was used for sampling the litter of forest stands was similar to the method used for herbaceous and shrubs (Karimian, 2009; Ghanbari, 2014).

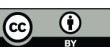



To study soil characteristics in grass and shrub cover, profile sampling point had been drilled in
the area under the canopy. Because microorganisms in the depth of 30 cm of the soil have the
highest deployment, and also the development of studied plant roots were mainly in this depth
too, therefore sampling carried out in each profile in the depth of 0-30 cm, and finally a sample
which weights around 1 kg was sent to the laboratory. Sampling in forest cover profiles, based
on the depth of soil, was usually done in the depth of 0-30 cm. Usually the dimensions of each
profile on the surface was $30 \times 30$ square cm and its depth was about 30 cm (or where the
bedrock was encountered). In forest covers, both from under the trees and the space between the
trees, soil sampling was done (Mahdavi *et al.*, 2007). Also in other land use-cover classes where
the structure did not include any plants, due to the presence of microorganisms in the soil, soil
sampling was done.
Samples of shoots, underground and litter plants were dried in the open air for a few weeks and
then their dry weight were determined using the scale (Karimian, 2009). Also, the wet samples
were placed in the oven at 72 ° C for 48 hours to completely dry, then dry weight of each sample
was calculated (Gholami, 2008). For determination of organic carbon of samples the combustion
in the furnace method was used (McDicken, 1997). On this basis, samples were grounded and 2
grams of each sample were separated. The samples were placed in the electric furnace for 24
hours at 375 ° C (Karimian, 2009) or 550 ° C for five hours (Ghanbari, 2014). After weighing the
ashes we took out of the electric furnace, the difference between initial weight and weight of the
ashes showed the amount of organic matter in the sample. With having the amount of organic
matter and by using the number 1 equation (Birdsey *et al.*, 2000), the amount of organic carbon
in each of the samples was calculated separately.
$$OC = \frac{1}{2} OM \qquad \text{Eq. 1}$$

Whereas, $OC$ is amount of organic carbon in gram and $OM$ is amount of organic matter in gram.
According to Eq. 1, half of the organic matter of the plants consists of organic carbon.
Soil samples, were then dried in the open air, crushed and sieved by 2 mm sieve and then some
experiments were conducted to determine the amount of organic matter and organic carbon
according to Walleky and Black method (Allison, 1965) and based on soil analysis methods. The
amount of soil organic matter was determined by measuring the amount of oxide-able organic
matter in the soil, therefore, after determining the percentage of organic carbon, the amount of
soil organic matter and soil organic carbon were calculated. Following the Walleky and Black
method, we poured one gram of soil in the Erlenmeyer flask, and then we poured 10 cc of normal
potassium dichromate and 20 cc of sulfuric acid into the Erlenmeyer flask and gently shake the
mix. After cooling the contents of the Erlenmeyer flask, we poured it into the volumetric flask
and then we brought the volume to 200 cc with distilled water. Then we added 10 drops of
orthophenanithroline to the solution and continued the titration by ferrous ammonium sulfate,
and we continued until the mixture became puce. At this time we read the volume of consumed
orthophenanithroline. The amount of soil organic matter was determined by measuring the
amount of oxide-able organic matter in the soil. Percentage of soil organic matter can be
calculated by Eq.2 (Haghighi, 2003; Gholami, 2008).
$$\%OM = \frac{(V_2 - V_1) \times N \times 0.39}{S} \qquad \text{Eq. 2}$$

Whereas, $V_1$ is amount of used ferrous ammonium sulfate for control sample in mL, $V_2$ is amount
of used ferrous ammonium sulfate for sample in mL, $N$ is Normality of ferrous ammonium
sulfate, $S$ is weight of dried soil sample in the open air and $OM$ is amount of organic material of



soil sample. Then by using Eq.3, the percentage of organic matter was converted into the percentage of organic carbon (Haghighi, 2003).

$$\%OC = \frac{\%OM}{1.724} \qquad \text{Eq. 3}$$

Whereas, $OC$ shows organic carbon content. Finally, by using the Eq.4, amount of organic carbon content of the soil was determined in kilograms per hectare (Haghighi, 2003).

$$OC = 10000 \times \%OC \times Bd \times E \qquad \text{Eq. 4}$$

Whereas, $Bd$ is density of soil sample in gram per cubic centimeter and $E$ is depth of sampling point in cm.

For simulating the land use-cover changes, the DINAMICA EGO software was used. All data used, before entering the simulation, were preprocessed in other applications such as ArcGIS, ENVI, IDRISI and all the images were studied in respect to their number of rows and columns, geographic coordinate system and overlapping border area and in different images were reviewed and modified (Makhdoom *et al.*, 2001).

In the simulation method used which was based on cellular automata method, the main rule amongst the cells was the possibility of any of the uses changing to another. To calculate the probability values, the Markov chain method was used (Soares-filho *et al.*, 2012). Then all the predictable variables were introduced to the model in form of auxiliary data. Before running the simulation based on the proposed model, all the proposed variables were examined in terms of their effectiveness on the model. First, for all of the modes of transition, the proposed variables were introduced as effective variables. After running the model and calculating effective variables coefficients matrix, the results were reviewed and for any transition, the variables that have been identified as ineffective and correlated were removed from the transition (Soares-filho *et al.*, 2009; KwadwoNti, 2013).

In the DINAMICA EGO software, land use-cover changes model was executed for a specified time period by using transition matrix, the effective variables coefficients matrix, effective variables values and the initial land use-cover status. The output of this model was simulated results of the area from $t_1$ timestamp to the specified time as the number of repetitions of the process. The time unit in the model was proportional to the purpose of the research which in most studies of land use-cover changes, yearly time unit is considered (KwadwoNti, 2013; Soares-filho *et al.*, 2009).

To validate the results of simulation, there are some tools in the DINAMICA EGO software which opens up a window with odd dimensions and moves it on the same position of both maps, and proportional to the size of each window, the minimum similarity is calculated on the basis of the fuzzy reciprocal similarity. After identifying the final simulation models of land use-cover changes, and using the final model, data on the current time entered into the model and the simulation process for the future was conducted (Soares-filho *et al.*, 2009; Soares-filho *et al.*, 2012; KwadwoNti, 2013).

Considering the measured data about the amount of carbon sequestration for each land use-cover class in the region which were measured by field studies and introduced laboratory experiments, the rate of carbon sequestration and its variations in accordance with the land use-cover changes was simulated for the future. For this purpose, if the region has $n$ number of land use-cover classes, and each class has $C_i$ number of cells, each class has carbon sequestration of $\overline{CS_i}$ ,the area of each cell on the ground is $S$, then the total amount of carbon sequestration in the region of the study can be calculated using Eq. 5.

$$CST = \sum_{i=1}^{n}(\overline{CS_i} \times C_i \times S) \qquad \text{Eq. 5}$$





In the Eq.5 for each $i = 1,2,...,n$, part   of $(\overline{CS}_i \times C_i \times S)$ shows the amount of carbon
sequestration in the $i^{th}$ class. The amount of carbon sequestration on each class can include
amount of carbon sequestration of the shoots of vegetation cover, underground organs, all of the
vegetation cover, soil and the amount of total average for the land use-cover class.
**Results**
The mean values for each class were determined as an indicator of carbon sequestration of the
soil in that class and the results are shown in the first column of Table 1. Carbon sequestrations
of plant samples were measured and according to the average volume of plants in each plot,
amount of carbon sequestration of plants per plot and eventually per hectare were found. After
the calculation for each sample in same the class, mean values were calculated for each land use-
cover class and were introduced as an indicator of the average sequestration plant of that class.
The results are presented in the second column of Table 1. Finally, the average of values of the
indicators of soil and plant for each land use-cover class were calculated and were introduced as
an indicator of total average for each class. The results are shown in the third column of Table 1.
Table 1 Mean values of carbon sequestration in land use-cover classes

| Land use-cover class | Mean values of carbon sequestration of soil (ton per Hectare) | Mean values of carbon sequestration in vegetation (ton per Hectare) | Mean values of total carbon sequestration (ton per Hectare) |
|---|---|---|---|
| Agriculture | 1178 | 28 | 1206 |
| Rangeland | 686 | 14 | 700 |
| Forest | 675 | 94 | 769 |
| Barren Area | 92 | -- | 92 |

In order to run simulation of changes in land use-cover classes, transition matrix between two
land use-cover maps between the years 1985 and 2000 was calculated based on Markov chain
method and for a 15-year period. Then, due to transitions and proposed variables, effective
variables coefficients matrix were calculated and the impact of variables was examined in each
transition. Also, the correlation between the coefficients by using Uncertainty Information Joint
method was calculated and finally one of two correlated variables or the variables which didn't
have any impact in transition were excluded from the model and final effective variables
coefficients matrix were stored. Some of final effective variables show in Table 2.
Table 2 A summary of final effective variables

| Transition_From | Transition_To | Variable | Significant |
|---|---|---|---|
| Residential | Agriculture | Distance_to_Agriculture | 1 |
| Residential | Agriculture | Distance_to_Forest | 0 |
| Residential | Agriculture | Soil_Texture | 1 |
| Residential | Rangeland | Distance_to_Agriculture | 1 |
| Forest | Rangeland | Distance_to_Roads | 1 |
| Forest | Agriculture | Soil_Texture | 1 |
| Forest | Residential | Distance_to_Rivers | 0 |

By using DINAMICA EGO software, the model ran starting from 1985 for a 15-years period to
get simulated result for year 2000. Then, minimum similarity between observed and simulated
result was determined by using fuzzy reciprocal similarity method for the same area regions to



validate the model. By using the specified model, changes of land use-cover in the city of Karaj
were determined from 1985 to 2014 in 5-years periods and the result is being shown in Table 3
3        Table 3 Changes of land use-cover in the city of Karaj from 1985 to 2014 (in hectare)

| Year / Class | Residential | Agriculture | Rangeland | Forest | Barren Area |
|---|---|---|---|---|---|
| 1985 | 7534.53 | 7289.37 | 17138.61 | 1412.37 | 2202.75 |
| 2000 | 9267.21 | 7201.98 | 13968.36 | 531.81 | 4608.27 |
| 2014 | 16002.54 | 3632.85 | 9433.98 | 89.37 | 6418.89 |

By continuing the process, the result of predictable changes of land use-cover was gained from
2014 to 2029 in the same way (Table 4).
Table 4 Predicted changes of land use-cover in the city of Karaj from 2019 to 2029 (in hectare)

| Year / Class | Residential | Agriculture | Rangeland | Forest | Barren Area |
|---|---|---|---|---|---|
| 2019 | 18240.89 | 2659.06 | 9916.14 | 73.29 | 4459.79 |
| 2024 | 19141.89 | 1830.01 | 9851.92 | 166.24 | 4345.46 |
| 2029 | 19705.01 | 1407.71 | 9657.84 | 193.10 | 4373.93 |

By combining the obtained results, the results of the simulation of carbon sequestration using the
simulation approach for land use-cover changes in the city of Karaj and Eq. 5 get calculated and
results are shown in Table 5. Residential class due to lack of impact on the process of carbon
sequestration were removed from the results.
11        Table 5 the results of the simulation of carbon sequestration from 2014 to 2029 (in ton)

| Year / Class | Agriculture | Rangeland | Forest | Barren Area | Total |
|---|---|---|---|---|---|
| 1985 | 43950039 | 6003740 | 54329 | 202719 | 50210827 |
| 2000 | 43423135 | 4893186 | 204570 | 424099 | 48944990 |
| 2014 | 21903662 | 3304770 | 34377 | 590730 | 25833539 |
| 2019 | 16032358 | 3473673 | 28192 | 410434 | 19944657 |
| 2024 | 11033739 | 3451176 | 63947 | 399912 | 14948774 |
| 2029 | 8487552 | 3383189 | 74279 | 402532 | 12347552 |

**Conclusion**
The results of this study corresponded with the findings of Soares-filho *et al.* (2012) that used
DINAMICA EGO software to study carbon sequestration in the agricultural land and low-
yielding forests of Brazil. White and Engelen (1993), Deadman *et al.* (1993), Itami (1994, 1988),
White and Engelen (1997), White (1998), Li and Yeh (2002), Barredo *et al.* (2003, 2004), Al-
Ahmadi *et al.* (2008), Al-Ghamdi (2012) and KwadwoNti (2013) also used the cellular automata
approach in studying the land use-cover changes and their results confirms the suitability of this
approach in the studies such as this.
In this study, by using fuzzy reciprocal similarity method, structural similarities between the
obtained data and observed data, in regions with an area of 900 to 26100 square meters, were
examined. The results show that the similarity between the observed and simulated results in a
15-year period increases with an increase in the size of compared area. Also, we can claim
results of simulation will catch up with reality for regions with an area of 900 square meters with
the probability of at least 30% and for regions with an area of 26100 square meters with a
probability of at least 70%. Sheng *et al.* (2012), Soares-fihlo *et al.* (2012, 2013) and KwadwoNti
(2013) also used fuzzy analysis method in validation of simulation results for examining land
use-cover changes and the results obtained by them validates the efficiency of methods based on
fuzzy analysis in the validation of the results of the simulation.
According to the simulations carried out in the city of Karaj it was determined that in the 15-year
period from 2014 to 2029, In terms of the extent of land use-cover classes, residential areas with



growth of about 10%, agriculture with 5% reduction, forests with a slight increase of about 0.3%, barren area with a reduction of about 6% and rangeland is also faced with a little fluctuation are almost stable. Considering the results of carbon sequestration in each land use-cover class and combining them with the results of land use-cover changes in the studied area it was observed that generally during 2014 and 2029 we will face a reduction in the amount of carbon sequestration, especially in agriculture class. The reason for this is the increase of residential areas and their expansion into other classes and given that residential areas have no measurable impact on carbon sequestration therefore development of these areas increase the ineffective classes in carbon sequestration of the region. According to the results of this research it is recommended to city managers to take effective measures to increase residential green spaces with appropriate plant species in the study area in such a way that in the future the increasing carbon emissions don't cause a sharp drop in air quality.

**Acknowledgement**

We acknowledge University of Tehran and National University Malaysia (UKM), for their provision of resources and collaborative efforts. This research was supported by the LRGS grant 203/PKT/6720004, UKM: XX-11-2012, and XX-15-2012.

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
