# Peer review of "Simulating Carbon Sequestration using Cellular Automata and land use assessment; Case of: Karaj City, Iran"

_Solid Earth, 2017_

## Referee Comment (RC1) · Anonymous Referee #1 · 6 Apr 2017

Dear Author,

Your manuscript demonstrates the simulation of Carbon Sequestration under different land use using Cellular Automata, in Karaj City, Iran. The idea of your work is good and really interested but I think you should do more efforts in your manuscript to make it better.

My rapid comments about your manuscript are: you should improve the abstract, see line 18 and 19 you can rephrase these lines. The introduction is good but you can present more references in it. In the part of material and method, you should present a map that contains a location of your study area and samples point. Also, for me it not clears enough why you used two methods for measuring the organic carbon (Loss On

Ignition and WB), and by which method you presented the final calculation of carbon stock, or you want to compare between methods so it should be presented in the results and discussion. You told us that you used geographical and remote sensing programs such as ArGIS, Envi and IDRISI (what versions) , and you did not illustrate any maps for land use or land cover changes under the studied periods (past, present and future ) . Results part contains interested result but it should improve so much and it should enclose a concentrated discussion. Last of all, the initial paragraph of conclusion it should not be there, and the conclusion should be focused, and some part of it can go to the part of results and discussions.

Finally, I would like to review your manuscript after you doing the suggested improvement.

All the best my colleague

---

## Short Comment (SC1) · 18 Apr 2017

1. The extent of the study area is not mentioned. 2. Why that City was chosen is missing 3. Description of the vegetation types and agricultural products is missing 4. Land use maps for the modelling are not described (scale, or basis of classification) 5. It only uses 4 explication variables ( distances to roads, distances to rivers, distances to agriculture and soil texture). This model could include population, population density, yields of crops, incomes from crops) in order to understand the land use change process in the area.. 6. the assumption is that at least the roads will be the same in the future and it is not explained. 7. Table 5 does not includes the area in which is distributed that Carbon. The units should be verified and please if they are right reported

in millions of tons. 8. The validation of the models is poorly explained and it is not possible to see how well the model performed.

---

## Short Comment (SC2) · 4 May 2017

The article represent a fair contribution to scientific progress within the scope of this journal. The scientific approach and applied methods should be explained better, the validation of the model is missing. The explanatory variables that were used are not exhaustive, so they need to explain why these variables were selected. Also there is a need to justify why this site was chosen, to contextualize the importance of the work. The scientific results and conclusions presented are in general clear, but there are some further work to be done, for example, in the units of carbon. The manuscript should be accepted after major revision. I would NOT be willing to review the revised paper because sadly I do not have time now. But I hope this work could be improved

and published.

---

## Referee Comment (RC2) · Anonymous Referee #2 · 29 May 2017

1 **Simulating Carbon Sequestration using Cellular Automata and land use assessment; Case**
2 **of: Karaj City, Iran**

5 **Article Type**: Research Paper

7 Ali KHATIBI
8 MSc. Environmental Science
9 Department of Environmental Sciences, Faculty of Natural Resources, College of Agriculture &
10 Natural Resources, University of Tehran, Karaj, Iran.

12 Sharareh POUREBRAHIM, Ph.D (corresponding author)
13 Dr., Assistant Professor
14 Department of Environmental Sciences, Faculty of Natural Resources, College of Agriculture &
15 Natural Resources, University of Tehran, Karaj, Iran. Tel: +989127617834
16 sh_pourebrahim@ ut.ac.ir
17

18 MAZLIN Bin Mokhtar, PhD [3]

19  Professor

[revised manuscript text omitted]

17 Table 4 Results of Markov Chain for the study area in 1985-2000

| Class | The most probability of transition | |
|---|---|---|
| | Value | Class name |
| Residential | 0.0094 | Rangeland |
| Agriculture | 0.0520 | Rangeland |
| Rangeland | 0.0307 | Barren Area |
| Forest | 0.1122 | Agriculture |
| Barren Area | 0.0604 | Residential |

18 Then, due to transitions and proposed variables, effective variables coefficients matrix were
19 calculated and the impact of variables was examined in each transition. Also, the correlation
20 between the coefficients by using Uncertainty Information Joint method was calculated and
21 finally one of two correlated variables or the variables which didn't have any impact in transition

1 were excluded from the model and final effective variables coefficients matrix were stored.
2 Some of final effective variables shows in Figure 4.

| Transition_From* | Transition_To* | Variable* | Ra... | Ra... | Pos... | Ex... | Weight_C... | Contrast | Significant |
|---|---|---|---|---|---|---|---|---|---|
| 1 | 2 | Distance_to_Agriculture/distance_to_2 | 0 | 100 | 23731 | 2312 | 1.052150... | 2.43332... | 1 |
| 1 | 2 | Distance_to_Agriculture/distance_to_2 | 100 | 200 | 17201 | 279 | -0.826841... | -0.9792... | 1 |
| 1 | 2 | Distance_to_Agriculture/distance_to_2 | 200 | 300 | 11364 | 98 | -1.466260... | -1.5942... | 1 |
| 1 | 2 | Distance_to_Agriculture/distance_to_2 | 300 | 400 | 7674 | 27 | -2.367915... | -2.4662... | 1 |
| 1 | 2 | Distance_to_Agriculture/distance_to_2 | 400 | 1800 | 17585 | 101 | -1.875604... | -2.1055... | 1 |
| 1 | 2 | Distance_to_Barrenarea/distance_to_5 | 0 | 100 | 7151 | 172 | -0.424850... | -0.4598... | 1 |
| 1 | 2 | Distance_to_Barrenarea/distance_to_5 | 100 | 200 | 7795 | 249 | -0.133003... | -0.1468... | 1 |
| 1 | 2 | Distance_to_Barrenarea/distance_to_5 | 200 | 300 | 7455 | 289 | 0.067639... | 0.07508... | 0 |
| 1 | 2 | Distance_to_Barrenarea/distance_to_5 | 300 | 6900 | 55154 | 2107 | 0.052402... | 0.19347... | 1 |
| 1 | 2 | Distance_to_Forest/distance_to_4 | 0 | 100 | 4971 | 381 | 0.789480... | 0.87141... | 1 |
| 1 | 2 | Distance_to_Forest/distance_to_4 | 100 | 200 | 8079 | 333 | 0.131526... | 0.14791... | 1 |
| 1 | 2 | Distance_to_Forest/distance_to_4 | 200 | 700 | 34322 | 1171 | -0.064898... | -0.1137... | 1 |
| 1 | 2 | Distance_to_Forest/distance_to_4 | 700 | 800 | 4800 | 83 | -0.761771... | -0.7970... | 1 |
| 1 | 2 | Distance_to_Forest/distance_to_4 | 800 | 900 | 4113 | 123 | -0.201045... | -0.2112... | 1 |
| 1 | 2 | Distance_to_Forest/distance_to_4 | 900 | 1000 | 3968 | 159 | 0.102098... | 0.10788... | 0 |
| 1 | 2 | Distance_to_Forest/distance_to_4 | 1000 | 1100 | 3105 | 76 | -0.406938... | -0.4209... | 1 |
| 1 | 2 | Distance_to_Forest/distance_to_4 | 1100 | 1600 | 10917 | 328 | -0.196241... | -0.2252... | 1 |
| 1 | 2 | Distance_to_Forest/distance_to_4 | 1600 | 1700 | 876 | 22 | -0.380572... | -0.3842... | 0 |
| 1 | 2 | Distance_to_Forest/distance_to_4 | 1700 | 1800 | 576 | 33 | 0.477714... | 0.48220... | 1 |
| 1 | 2 | Distance_to_Forest/distance_to_4 | 1800 | 1900 | 372 | 34 | 0.981630... | 0.98924... | 1 |
| 1 | 2 | Distance_to_Forest/distance_to_4 | 1900 | 2300 | 1084 | 72 | 0.635298... | 0.64755... | 1 |
| 1 | 2 | Distance_to_Forest/distance_to_4 | 2300 | 2400 | 109 | 0 | 0 | 0 | 0 |
| 1 | 2 | Distance_to_Forest/distance_to_4 | 2400 | 5600 | 263 | 2 | -1.593057... | -1.5958... | |

4 ==Figure== 4 A summary of final effective variables
5 By using DINAMICA EGO software, the model ran starting from 1985 for a 15-years period to
6 get simulated result for year 2000. Then, minimum similarity between observed and simulated
7 result was determined by using fuzzy reciprocal similarity method for the same area regions to
8 validate the model (Table. 5).
9 Table 5 the minimum similarity between observed and simulated result in year 2000

| Area (square meters) | Minimum Similarity |
|---|---|
| 900 | 0.30 |
| 4500 | 0.41 |
| 8100 | 0.50 |
| 15300 | 0.61 |
| 26100 | 0.70 |

10 Finally, the simulation process for the city of Karaj was carried out in a period of 15 years
11 starting from 2014, and the results obtained are presented in Table 6. Also, the results have been
12 presented in Figure 5-7.

[Figure]

Figure 5 Simulated land use-cover classes in 2019

Figure 6 Simulated land use-cover classes in 2024

[Figure]

Figure 7 Simulated land use-cover classes in 2029

Table 6 Changes of land use-cover in the city of Karaj from 2014 to 2029 (in hectare)

| Year / Class | Residential | Agriculture | Rangeland | Forest | Barren Area |
|---|---|---|---|---|---|
| 1985 | 7534.53 | 7289.37 | 17138.61 | 1412.37 | 2202.75 |
| 2000 | 9267.21 | 7201.98 | 13968.36 | 531.81 | 4608.27 |
| 2014 | 16002.54 | 3632.85 | 9433.98 | 89.37 | 6418.89 |
| 2019 | 18240.89 | 2659.06 | 9916.14 | 73.29 | 4459.79 |
| 2024 | 19141.89 | 1830.01 | 9851.92 | 166.24 | 4345.46 |
| 2029 | 19705.01 | 1407.71 | 9657.84 | 193.10 | 4373.93 |

By combining the obtained results, the results of the simulation of carbon sequestration using the simulation approach for land use-cover changes in the city of Karaj and equation number 5 get calculated and results are shown in Table 7 and Figures 8 and 9. Residential class due to lack of impact on the process of carbon sequestration were removed from the results.

Table 7 the results of the simulation of carbon sequestration of the study area from 2014 to 2029 (in ton)

| Year / Class | Agriculture | Rangeland | Forest | Barren Area | Total |
|---|---|---|---|---|---|
| 1985 | 43950039 | 6003740 | 54329 | 202719 | 50210827 |
| 2000 | 43423135 | 4893186 | 204570 | 424099 | 48944990 |
| 2014 | 21903662 | 3304770 | 34377 | 590730 | 25833539 |
| 2019 | 16032358 | 3473673 | 28192 | 410434 | 19944657 |
| 2024 | 11033739 | 3451176 | 63947 | 399912 | 14948774 |
| 2029 | 8487552 | 3383189 | 74279 | 402532 | 12347552 |

[Figure]

Figure 8 Status of carbon sequestration amount in the land use-cover classes of the study area in 1985-2029

[Figure]

Figure 9 Changes of total amount of carbon sequestration of the study area in 1985-2029

[revised manuscript text omitted]

---

## Author Comment (AC1) · 30 Jun 2017

First of all, we would like to thanks for very great comments of reviewer (1) to help us improve the manuscript and to inform you that the revised version of our manuscript with considering all comments of reviewer (1) has been prepared and it is ready for submission to your journal. Based on comments, the outline of corrections is as follows: You should improve the abstract, see line 18 and 19 you can rephrase these lines. The abstract revised based on your comment. It starts out with one line introduction, the study objective, the methodology, main finding and conclusion. Line 18 and 19 revised. These changes have been highlighted in the text.

[Figure]

The introduction is good but you can present more references in it. Some references has been added to this part and highlighted in the text

In the part of material and method, you should present a map that contains a location of your study area and samples point. About half page has been added to the methodology to explain study area and the reason of choosing this location. It is followed by the Figure 1 to show the location of Karaj city in Iran.

For me it not clears enough why you used two methods for measuring the organic carbon (Loss On C1 Ignition and WB), and by which method you presented the final calculation of carbon stock, or you want to compare between methods so it should be presented in the results and discussion. Two methods for measuring sequestrated carbon that we used in the study are different base on type of samples. For plant samples, it was used plant dry weight method and for soil samples we used Walleky and Black method. Finally the mean values obtained by each method used to calculate for each land use-cover class. This explanation has been added into the second paragraph of methodology and has been highlighted in the text.

You told us that you used geographical and remote sensing programs such as ArGIS, Envi and IDRISI (what versions) , and you did not illustrate any maps for land use or land cover changes under the studied periods (past, present and future ) . Figure 3-7 has been added to show the land use/cover changes.

Results part contains interested result but it should improve so much and it should enclose a concentrated discussion. Discussion (one page) has been added at the end of the results.

Last of all, the initial paragraph of conclusion it should not be there, and the conclusion should be focused, and some part of it can go to the part of results and discussions. Based on your great comments, conclusion has been revised and rephrased. The changes has been highlighted in the text.

Sincerely, The Authors

Please also note the supplement to this comment:
https://www.solid-earth-discuss.net/se-2017-20/se-2017-20-AC1-supplement.pdf

———————————————

[Figure]

**Supplement:**

**Simulating Carbon Sequestration using Cellular Automata and land use assessment; Case of: Karaj City, Iran**

**Abstract**
Carbon sequestration has been proposed as a way to slow the atmospheric and marine accumulation of greenhouse gases, which are released by burning fossil fuels and measuring it is important to predict environmental conditions both in atmosphere and terrestrial ecosystems. In this study, land use-cover changes and their simulation status has been used for predicting and investigating the rate of carbon sequestration. The Karaj city as one of the metropolises of Iran was selected as study area. The nature of Karaj city has changed from garden with agriculture performance to an inharmonic city with rapid population. High quality agricultural, green spaces and gardens have changed to the industrial, settlements and urban services very fast. Five classes of land use/cover including residential, agriculture, rangeland, forest and barren areas were considered and randomly in each class a total of 20 points were selected and vegetation and soil samples were taken. In plant samples, the amount of carbon sequestration was determined by calculating the amount of organic carbon by plant dry weight method and for determining the amount of carbon sequestration in soil samples we used Walleky and Black method, too. For each area, the average value of carbon sequestration of plant and soil samples was introduced as 'carbon sequestration index' of that class. Average values for each class were determined as an indicator of carbon sequestration of that class and then by using the DINAMICA EGO software a simulation was conducted using cellular automata approach to simulate changes in the classes of land use/cover in the city of Karaj. Finally, by using carbon sequestration index and the results of the simulation, changes in carbon sequestration in each class were calculated. On this basis, it was found that in the 15-year period from 2014 to 2029, agricultural land use with the greatest amount of carbon sequestration will be faced with the huge reduction, because of expansion of the residential area. The results show that this method can be used as an appropriate approach for similar studies.
**Keywords:** Carbon Sequestration, Land Use-Cover Changes, Cellular Automata, City of Karaj

**Introduction**
Carbon sequestration can be defined as the process of removing carbon from the atmosphere and depositing it in a reservoir (UNFCCC, 2015). It has been proposed as a way to slow the atmospheric and marine accumulation of greenhouse gases, which are released by burning fossil fuels (Hodrien, 2008). Terrestrial carbon sequestration is the result of a balance between the different stages of the carbon cycle in the biosphere and pedosphere, such as photosynthesis, plant growth, congestion and carbon accumulation in soils and carbon emissions from breathing organisms, microbial decomposition of leaf litter, and oxidation of organic carbon in soil and land degradation. Several factors are involved in this process, which are classified in two categories of physical and managerial factors. Physical factors are divided into three categories of factors: soil, climate and terrain geometry. These factors are difficult and sometimes impossible to control. Therefore, in the management of carbon sequestration, only managerial factors are in the control of humans (Ferreiro *et al.*, 2010).
Because the majority of carbon sequestration is in the soil, organic carbon of the soil plays an important role in the process of carbon sequestration. The amount of carbon in the soil significantly changes according to location, topography, bedrock or vegetation type and previous management approaches. Also time-wise, in the growing season and the time of decomposition

processes, the amount of carbon found in the roots, litter and biomass of microorganisms in the soil can vary. Instead, processes that reduce the amount of organic matter in soil include erosion, compaction and reduction of soil permeability, loss of soil structure, mineralization and oxidation of humid substances (Bruce *et al.*, 1999).

Although the amount and rate of carbon sequestration is higher in mild and humid tropical forest ecosystems however, the speed of the chemical analysis and biological processes that cause the release of carbon dioxide, due to high humidity and high ambient temperature is also very high. Therefore, the increase in biomass of hard wood plants in arid and semi-arid regions has many advantages due to the reduction of carbon sequestration (Lal, 2008). That is why international organizations such as FAO and UNDP have chosen these regions to implement programs of carbon sequestration to reduce greenhouse gases (Ghanbari, 2014).

Land use change is a major driver of terrestrial ecosystem carbon storage (Chuai et al., 2013). Monitoring and modeling of historical situation can be used to anticipate the negative effects of these changes in land use-cover, in order to protect resources, effectively (Şatır & Berberoğlu, 2012). One of the most popular methods to simulate spatial changes is cellular automata. A cellular automata is a system of a finite number of cells that are located next to each other in terms of location and continuity and are stored together in a regular grid of cells in the form of a raster image (KwadwoNti, 2013). By this way, the target land use-cover is divided into multiple grid classes as cells. This classification includes sorting the objects in groups or collections of objects based on the relationships that exist between them (Sokal, 1974; Anderson *et al.*, 1976; Gregorio & Jansen, 2005; Şatır & Berberoğlu, 2012). Finally by using cellular automata method, the changes of land use-cover classes simulate in a specific time period. This model is well suited to simulate phenomenon of spatial landscape. (Al-ghamdi, 2012; Guan & Clarke, 2010; Jantz *et al.*, 2010; KwadwoNti, 2013. White and Engelen, 1993, Deadman *et al.,* 1993, Itami, 1994, , White and Engelen, 1997; White, 1998; Li and Yeh, 2002; Barredo *et al.,* 2003, 2004; Al-Ahmadi *et al.,* 2008.

**Materials and Methods**

The Karaj city as one of the metropolises of Iran was selected as study area. This city has been faced with a rapid growth rate in population and settlement areas since the last two decades (MRUD, 2012). The nature of Karaj city has changed from garden with agriculture performance to an inharmonic city with rapid population. High quality agricultural, green spaces and gardens have changed to the industrial, settlements and urban services very fast.It is located in $35^{\circ}42'N$, $50^{\circ}50'E$ to $35^{\circ}53'N$, $51^{\circ}03'E$ (Fig. 1). Five classes of land use/cover including residential, agriculture, rangeland, forest and barren areas were considered and randomly in each class a total of 20 points were selected and vegetation and soil samples were taken. (Fig. 2).

In this study, we used two types of samples, plant and soil, and two different methods were used to determine the amount of carbon sequestration in each type. In plant samples, it was done by plant dry weight method and in soil samples we used Walleky and Black method. Sampling at any point was usually done in a season when plant growth in that area had been completed. Which means, depending on the coverage of the area, and depending on the type of plant species in their sampling time was determined (Karimian, 2009). For samples of grass and shrub cover, it was sampled the above ground and underground biomasses separately. For measuring and sampling the above ground biomass, it was used a direct measurement method. For this purpose, the above ground parts of biomass was cut completely and after drying them in the laboratory,

weighed, and then by including plant basis weight per unit area, weight of above ground biomass and total biomass per hectare were estimated (Mesdaghi, 2001).

[Figure]

Figure 1 The location of Karaj city in Iran

[Figure]

Figure 2 The location of sampling points in Karaj city

In the sampling points, plots with specific measures (1 × 1 square m) had been placed. Litter in each plot was harvested as part of the above ground biomass. Litter samples were transported to the laboratory and after getting fully dried, the dry weight of each sample was recorded separately and then the total weight of litter per square meter was determined. Also,

belowground biomass of the sampled species were extracted. Usually these profiles had the dimensions of $30 \times 30 \times 30$ cubic cm. In each plot, the roots were collected and after drying, weighted, and then by calculating the root weight per unit area the amount of studied factors were determined. In forest cover, the tree species were selected with at least 10 cm diameters to avoid selecting of very young samples. In order to determine the weight of biomass of standing trees, depending on forest density and homogeneity of mass, plots of different sizes were used. To do this, plots were randomly located in forest stand and the numbers of the tree in each were counted and then we attempted to measure the height and diameter at 1.5 meter height to determine the volume of the tree. To determine the bulk density of the tree, we took samples from a tree branch in each stand. For this purpose, a piece of tree branches with specified dimensions were cut and then their dry weight and volume were measured. By dividing weight by volume, bulk density of the tree was determined. Then, by multiplying the volume of trees in bulk density, biomass weight per unit area was determined. The method was used for sampling the litter of forest stands was similar to the method used for herbaceous and shrubs (Karimian, 2009; Ghanbari, 2014).

To study soil characteristics in grass and shrub cover, profile sampling point had been drilled in the area under the canopy. Because microorganisms in the depth of 30 cm of the soil have the highest deployment, and also the development of studied plant roots were mainly in this depth too, therefore sampling carried out in each profile in the depth of 0-30 cm, and finally a sample which weights around 1 kg was sent to the laboratory. Sampling in forest cover profiles, based on the depth of soil, was usually done in the depth of 0-30 cm. Usually the dimensions of each profile on the surface was $30 \times 30$ square cm and its depth was about 30 cm (or where the bedrock was encountered). In forest covers, both from under the trees and the space between the trees, soil sampling was done (Mahdavi *et al.*, 2007). Also in other land use-cover classes that the structure did not include any plants such as some barren areas, only soil samples were taken.

[revised manuscript text omitted]

As we mentioned, two types of samples were taken to measure the amount of carbon sequestration, including soil and plant samples. After measuring the amount of the sequestered carbon in each hectare, the mean values for each class were determined as an indicator of carbon sequestration of the soil in that class by using Walleky and Black method and the results are shown in Table 1.

Table 1 Mean values of carbon sequestration of soil per land use-cover classes using Walleky and Black method

| Land use-cover class | Mean values ($\pm 0.01$) of carbon sequestration in soil (millions of tons per Hectare) |
|---|---|
| Agriculture | $11.78 \times 10^{-4}$ |
| Rangeland | $6.86 \times 10^{-4}$ |
| Forest | $6.75 \times 10^{-4}$ |
| Barren Area | $0.92 \times 10^{-4}$ |

After the calculation of soil tests, carbon sequestrations of plant samples were measured and according to the average volume of plants in each plot, amount of carbon sequestration of plants per plot and eventually per hectare were found. After the calculation for each sample in same the class by using plant dry weight method, mean values were calculated for each land use-cover class and were introduced as an indicator of the average sequestration plant of that class (Table 2). In the study area, there are some different types of vegetation. For example, in agriculture points most of plants were Jat, Alfalfa and potatoes. In rangeland points, most of samples were included of Sagebrush and in the forest points, Plane tree and Cedar were the most important plants that were sampled.

Table 2 Mean value of carbon sequestration in vegetation per land use-cover classes using plant dry weight method

| Land use-cover class | Mean values ($\pm$ 0.01) of carbon sequestration in vegetation (millions of tons per Hectare) |
|---|---|
| Agriculture | $0.28 \times 10^{-4}$ |
| Rangeland | $0.14 \times 10^{-4}$ |
| Forest | $0.94 \times 10^{-4}$ |

Finally, the mean values of the indicators of soil and plant for each land use-cover class (two previous tables) were calculated and were introduced as a total indicator for each class. The results are shown in Table 3.

Table 3 Mean amount of total carbon sequestration per land use-cover classes

| Land use-cover class | Mean values ($\pm$ 0.01) of carbon sequestration (millions of tons per Hectare) |
|---|---|
| Agriculture | $12.06 \times 10^{-4}$ |
| Rangeland | $7.00 \times 10^{-4}$ |
| Forest | $7.69 \times 10^{-4}$ |
| Barren Area | $0.92 \times 10^{-4}$ |
| Total | $27.67 \times 10^{-4}$ |

In order to run simulation of changes in land use-cover classes, transition matrix between two land use-cover maps between the years 1985 and 2000 was calculated based on Markov chain method and for a 15-year period (Table 4).

Table 4 Results of Markov Chain for the study area in 1985-2000

| Class | The most probability of transition | |
|---|---|---|
| | Value | Class name |
| Residential | 0.0094 | Rangeland |
| Agriculture | 0.0520 | Rangeland |
| Rangeland | 0.0307 | Barren Area |
| Forest | 0.1122 | Agriculture |
| Barren Area | 0.0604 | Residential |

Then, due to transitions and proposed variables, effective variables coefficients matrix were calculated and the impact of variables was examined in each transition. Also, the correlation between the coefficients by using Uncertainty Information Joint method was calculated and finally one of two correlated variables or the variables which didn't have any impact in transition were excluded from the model and final effective variables coefficients matrix were stored. Some of final effective variables shows in Table 5.

Table 5 A summary of final effective variables

| Transition_From | Transition_To | Variable | Significant |
|---|---|---|---|
| Residential | Agriculture | Distance_to_Agriculture | 1 |
| Residential | Agriculture | Distance_to_Forest | 0 |

| Residential | Agriculture | Soil_Texture | 1 |
|---|---|---|---|
| Residential | Rangeland | Distance_to_Agriculture | 1 |
| Forest | Rangeland | Distance_to_Roads | 1 |
| Forest | Agriculture | Soil_Texture | 1 |
| Forest | Residential | Distance_to_Rivers | 0 |
| Forest | Residential | Distance_to_Roads | 0 |

By using DINAMICA EGO software, the model ran starting from 1985 for a 15-years period to get simulated result for year 2000 (Figs. 3 & 4). Then, minimum similarity between observed and simulated result was determined by using fuzzy reciprocal similarity method for the same area regions to validate the model (Table 6).

Table 6 the minimum similarity between observed and simulated result in year 2000

| Area (square meters) | Minimum Similarity |
|---|---|
| 900 | 0.30 |
| 4500 | 0.41 |
| 8100 | 0.50 |
| 15300 | 0.61 |
| 26100 | 0.70 |

Finally, the simulation process for the city of Karaj was carried out in a period of 15 years starting from 2014, and the results have been presented in Figures 5-7.

[Figure]

| Figure 3 Observed land use-cover classes in 1985 | Figure 4 Simulated land use-cover classes in 2000 |
|---|---|

[Figure]

Figure 5 Simulated land use-cover classes in 2019

Figure 6 Simulated land use-cover classes in 2024

Figure 7 Simulated land use-cover classes in 2029

In the other words, by using the specified model, changes of land use-cover in the city of Karaj were determined from 1985 to 2014 in 5-years periods and the result is being shown in Table 7.

Table 7 Changes of land use-cover in the city of Karaj from 1985 to 2014 (in hectare)

| Year / Class | Residential | Agriculture | Rangeland | Forest | Barren Area |
|---|---|---|---|---|---|
| 1985 | 7534.53 | 7289.37 | 17138.61 | 1412.37 | 2202.75 |
| 2000 | 9267.21 | 7201.98 | 13968.36 | 531.81 | 4608.27 |
| 2014 | 16002.54 | 3632.85 | 9433.98 | 89.37 | 6418.89 |

As it can be seeing in Table 7, the results showed the past changes in the study area. By continuing the process, the result of predictable changes of land use-cover was gained from 2014 to 2029 in the same way (Table 8).

Table 8 Predicted changes of land use-cover in the city of Karaj from 2019 to 2029 (in hectare)

| Year / Class | Residential | Agriculture | Rangeland | Forest | Barren Area |
|---|---|---|---|---|---|
| 2019 | 18240.89 | 2659.06 | 9916.14 | 73.29 | 4459.79 |
| 2024 | 19141.89 | 1830.01 | 9851.92 | 166.24 | 4345.46 |
| 2029 | 19705.01 | 1407.71 | 9657.84 | 193.10 | 4373.93 |

By combining the obtained results, the results of the simulation of carbon sequestration using the simulation approach for land use-cover changes in the city of Karaj and Eq. 5 get calculated and results are shown in Table 9. Residential class due to lack of impact on the process of carbon sequestration were removed from the results.

Table 9 the results of the simulation of carbon sequestration of the study area from 2014 to 2029 (millions of tons)

| Year / Class | Agriculture | Rangeland | Forest | Barren Area | Total |
|---|---|---|---|---|---|
| 1985 | 43.95 | 6.00 | 0.05 | 0.20 | 50.21 |
| 2000 | 43.42 | 4.89 | 0.20 | 0.42 | 48.94 |
| 2014 | 21.90 | 3.30 | 0.03 | 0.59 | 25.83 |
| 2019 | 16.03 | 3.47 | 0.02 | 0.41 | 19.94 |
| 2024 | 11.03 | 3.45 | 0.06 | 0.39 | 14.94 |
| 2029 | 8.48 | 3.38 | 0.07 | 0.40 | 12.34 |

It is very important to measure changes rate of many features. These rates, at least, help us to understand the behavior of phenomenon and predict the future of them (Webster, 2016) and finally we can use them to plan especially in using the environment. In this study, our main goal was to determine the rate of changes of carbon sequestration based on changes rate in land use-cover by time and answering to this question that if this approach is an appropriate method. So in two separates parts we indicated the amount of carbon sequestration in five land use-cover classes in lab and also we simulated the changes in those classes by using satellite images and cellular automata method. We selected City of Karaj as the study area which has been faced with a sharp growth rate in population and settlement areas since the last two decades. There are three reasons for this sharp growth. First, City of Karaj is near the Tehran and many daily job migrants have settled in City of Karaj. The second reason is City of Karaj becomed the capital of Alborz province in about 8 years ago and it would face to many types of developments especially in urban structures. The third reason to select this area was that the agriculture and industry were and are two major types of occupation in City of Karaj (Khatibi, 2016; Sakieh, 2013; MRUD, 2012). So it had been seem that this area was an appropriate place to both simulate the land use-cover changes and measuring the amount of carbon sequestration. According to the simulations carried out in the city of Karaj it was determined that in the 15-year period from 2014 to 2029, in terms of the extent of land use-cover classes, residential areas with growth of about 10%, agriculture with 5% reduction, forests with a slight increase of about 0.3%, barren area with a reduction of about 6% and rangeland is also faced with a little fluctuation are almost stable. Finally, considering the results of carbon sequestration in each land use-cover class and combining with the results of land use-cover changes in the studied area it was observed that generally during 2014 and 2029 we will face a reduction in the amount of carbon sequestration. The reason for this is the increase of residential areas and their expansion into other classes and

given that residential areas have no measurable impact on carbon sequestration therefore development of these areas increase the ineffective classes in carbon sequestration of the region. In this study we measured the amount of carbon sequestration for two types of samples, plant part and soil part for each sampling point and results showed in table 1 and 2. So, it was gotten an index for each class as a mean value that the results is in table 3. As we can see, the amount of carbon sequestration in soil samples of agriculture class was enormous high. Although, there is no single unifying volume that determine the impact of different land management practices on soil carbon sequestration in agricultural land use (World bank, 2012) and this truth that the amount of carbon sequestration has a varied changes in related to plant growth seasons (Forge, 2001), but it is still useful to determine and monitor its. We took our samples in April that is the beginning of plant growth season and most of farmers use many types of fertilizers, natural and chemical, to gain a better plant growth. So, it is clear that our amount of carbon sequestration was enormous high, especially right after fertilizing. In fact, we measured the amount of carbon in fertilizer added to soil. So, it could be removed those data and made an index only by using the plant part data, but we decided not to eliminate those data because our main goal in this study was investigating the ability of using land use-cover change simulation in predicting the status of carbon sequestration and on the other hand we wanted an index for total carbon sequestrated in the soil without its origin or cause.

For simulating of land use cover changes, it was used satellite images onto a cellular model to determine changes rate between the classes. This rate was modeled by using Markov Chain method using between initial and final categorical images of the study area and calculating the probability of all possible changes that it showed in table 4. As it can be seen the maximum probability is for changing of forest to agriculture and the minimum amount is for transiting residential to rangeland. Then, some other possible effective variables introduce to the model and their significances in model were determined (results in table 5) and the correlated and insignificant variables removed from the model. So, the model were run to simulate the status of land use-cover classes for next 15 years by using DINAMICA EGO. Finally, as it can be seen in table 6, we used fuzzy reciprocal similarity method that showed structural similarities between the obtained data and observed data in regions with an area of 900 to 26100 square meters. This results show that the similarity between the observed and simulated results in a 15-year period increases with an increase in the size of compared area. Also, we can claim results of simulation will catch up with reality for regions with an area of 900 square meters with the probability of at least 30% and for regions with an area of 26100 square meters with a probability of at least 70%.

**Conclusion**

In this study, we used the results of land use-cover changes and their simulation status for predicting the rate of carbon sequestration in a specific location and the results show that this method can be used as an appropriate approach for similar studies. As the result shows land use change has been reduce the amount of carbon sequestration. The results of this study corresponded with the findings of Soares-filho *et al.* (2012) that used DINAMICA EGO software to study carbon sequestration in the agricultural land and low-yielding forests of Brazil. Sheng *et al.* (2012), Soares-fihlo *et al.* (2012, 2013) and KwadwoNti (2013) also used fuzzy analysis method in validation of simulation results for examining land use-cover changes and the results obtained by them, as our results, validates the efficiency of methods based on fuzzy analysis in

the validation of the results of the simulation.Based on the result, and because of observing and deserving sharp changes in residential areas, it is recommended to city managers to take effective measures to increase residential green spaces with appropriate plant species in the study area in such a way that in the future the increasing carbon emissions don't cause a sharp drop in air quality. Also we recommend some similar studies will be done due to compare the results and answering to this question that if we can use this method for other locations as a global approach. Although, we used this method for a specific location but the results investigate the ability of combining of simulation of land use-cover changes and carbon sequestration rate. Therefore, we used some variables but it seems there are some other variables that can affect the results of the study such as population density, yields of crops and etc. Some of these variables such as population density is hidden in other variables like settlement growth that presented as two variables named settlement class and distance to settlement area that we used them in this study but it is recommended to use more other variables in future same studies and investigating of their effects on the results.

**Acknowledgement**

[revised manuscript text omitted]

World Bank. 2012. *Carbon Sequestration in Agricultural Soils*. Washington, DC. © World Bank. [Online] Available at: https://openknowledge.worldbank.org/handle/10986/11868 License: CC BY 3.0 IGO."

Xiaowei C., Xianjin H., Wanjing W., Changyan W., Rongqin Z. 2014. Spatial Simulation of Land Use based on Terrestrial Ecosystem Carbon Storage in Coastal Jiangsu, China